# The Development and Psychometric Properties of Malay Language Child Oral Health Impact Profile—Short Form 19 (ML COHIP-SF 19)

**DOI:** 10.3390/healthcare13030257

**Published:** 2025-01-28

**Authors:** Noor Rashidah Ismail, Su Keng Tan, Norashikin Abu Bakar, Noren Nor Hasmun

**Affiliations:** 1Centre of Paediatric Dentistry & Orthodontics Studies, Faculty of Dentistry, Universiti Teknologi MARA (UiTM), Sungai Buloh 47000, Malaysia; noorrashidah.ismail85@gmail.com (N.R.I.); norashikin8234@uitm.edu.my (N.A.B.); 2Centre for Oral & Maxillofacial Surgery Studies (OMFS), Faculty of Dentistry, Universiti Teknologi MARA (UiTM), Sungai Buloh 47000, Malaysia; tansukeng@uitm.edu.my; 3Department of Oral Sciences, Faculty of Dentistry, University of Otago, Dunedin 9054, New Zealand

**Keywords:** oral health, quality of life, COHIP-SF 19 questionnaire, child, translation, validation

## Abstract

**Background/Objectives:** The Child Oral Health Impact Profile—Short Form 19 (COHIP-SF 19) is widely used to measure the oral health-related quality of life (OHRQoL) of children and adolescents. The current study aimed to validate the Malay language version of the COHIP-SF19 (ML COHIP-SF 19) and to assess its psychometric properties among Malaysian children/adolescents. **Methods:** Children aged from 9 to 16 years attending the Faculty of Dentistry, Universiti Teknologi MARA (UiTM) participated in this study. The original English version of the COHIP-SF 19 was translated using forward- and back-translation. The psychometric properties of the final version were tested for reliability and validity using Cronbach’s alpha, a non-parametric Spearman’s correlation test, and confirmatory factor analysis (CFA). **Results:** A total of 252 children aged from 9 to 16 years (mean age = 11.33 ± 1.87 years) self-completed the ML-COHIP-SF 19. The total scores of the ML COHIP-SF 19 ranged from 20 to 75 (mean = 55.67 ± 10.45) with an internal consistency (α) of 0.81. Convergent validity showed a fair correlation between self-perceived oral health rating and total ML COHIP-SF 19 scores, as well as the socio-emotional well-being subscale scores (rs = 0.38–0.42, *p* < 0.01). **Conclusions:** The ML COHIP-SF 19 demonstrated reliable psychometric properties and acceptable four-factor model fits, indicating that it is a valid tool to measure the OHRQoL of Malaysian children aged from 9 to 16 years.

## 1. Introduction

Oral health-related quality of life (OHRQoL) is a multidimensional construct that measures the impact of oral diseases and disorders on valued aspects of daily life. This construct captures the extent to which these conditions, through their frequency, severity, or duration, influence an individual’s perception of life quality [1]. It incorporates general health and well-being in agreement with the true concept of health, where health is the state of complete physical, social, and mental well-being and not just the absence of disease [2]. A comprehensive understanding of OHRQoL is vital in problem identification and prioritization, communication, screening for hidden issues, collaborative clinical decision-making, and monitoring treatment changes or responses to treatment [3].

Various OHRQoL measurement tools have been developed to systematically measure children’s quality of life (QoL) across diverse oral health conditions, encompassing patient-reported outcome measures (PROMs). PROMS, such as the Early Childhood Oral Health Impact Scale (ECOHIS) [4], rely on parents or caregivers as proxies to report on the child’s OHRQoL. However, concerns have been raised about the reliability of proxy reports in accurately reflecting the child’s personal experiences. A systematic review found that valid and reliable information can be obtained directly from children when appropriate, age-specific measurement tools are used. However, a notable discrepancy often exists between parents’ and children’s ratings of the children’s OHRQoL, particularly in the social and emotional well-being domains [5]. Therefore, while proxy reports provide valuable insights, they should be considered as supplementary to the child’s own assessment [6].

This has led to the development of various self-reported OHRQoL measurement tools, including the Child Perceptions Questionnaire (CPQ) [7,8], Child Oral Impacts on Daily Performances (C-OIDP) [9], and Child Oral Health Impact Profile (COHIP) [10,11]. The COHIP was developed in the United States of America (USA) by Broder et al., 2007. The original version has 34 items and 5 subscales: oral health, functional well-being, social/emotional well-being, school environment, and self-image [10]. This questionnaire was revised into a shorter version, the Child Oral Health Impact Profile—Short Form 19 (COHIP-SF 19), where the weak loadings were removed and only 19 items remained [11]. The COHIP-SF 19 questionnaire contains three conceptual subscales: five items in oral health (OH), four items in functional well-being (FWB), and ten items in socio-emotional well-being (SEWB). Two items (Item 8 and Item 15) are positively worded questions that assess how confident and attractive the children perceive themselves to be, respectively. COHIP-SF 19 has been translated into various languages to accommodate the linguistic differences and cross-cultural considerations of distinct target populations in different countries. This includes Mandarin (the Chinese version of the COHIP-SF 19) [12], German (the COHIP-G19) [13], Arabic (the Arabic COHIP-SF 19) [14], Portuguese (the Brazilian version of the COHIP-SF19) [15], Japanese (the COHIP-SF 19 JP) [16], Myanmar (the Myanmar COHIP-SF 19) [17], French (the French COHIP-SF-19) [18], and Indonesian (the Indonesian version of the COHIP-SF 19) [19]. Even though the Indonesian version of the COHIP-SF 19 is available, this questionnaire might not be applicable to children in Malaysia due to differences in colloquialisms and pronunciation.

Most previous OHRQoL studies in Malaysia have primarily focused on preschool-aged children, resulting in limited knowledge regarding the impact of dental caries and pain on the quality of life of primary and secondary schoolchildren in Malaysia. Addressing this gap is essential as it will provide a more comprehensive understanding of Malaysian children’s oral health. At present, no research has explored the impact of dental pain and dental caries on the OHRQoL of children in Malaysia. A review of the literature suggested that there is currently no single questionnaire in the Malay language suitable for assessing subjective perceptions across a broad age range that includes both primary and secondary schoolchildren. Although the COHIP-SF 19 is considered a promising self-reported tool due to its applicability for a broad age range, no published studies have yet addressed its translation and validation into the Malay language. Thus, the present study aimed to translate and validate the Malay language version of the COHIP-SF 19 and assess its psychometric properties among Malaysian children.

## 2. Material and Methods

### 2.1. Study Design and Ethics Approval

This study consisted of two phases: Phase 1 focused on the cross-cultural adaption of the original English COHIP-SF 19 to the Malay language, while Phase 2 utilized the translated and validated questionnaire to evaluate the perceived impact of dental caries and dental pain on oral health-related quality of life (OHRQoL) of children in Malaysia. For Phase 2, dental caries was recorded using International Caries Detection and Assessment System (ICDAS) and dental pain was measured using Faces Pain Scales—Revised. However, this paper presents the findings from Phase 1 only.

This cross-sectional study was conducted at the Faculty of Dentistry, Universiti Teknologi MARA (UiTM), Malaysia among children aged from 9 to 16 years. Permission to translate the questionnaire, namely COHIP-SF 19, was obtained from the developer, Professor Hillary Broder. The study protocol was reviewed and approved by the Research Ethics Committee, Research Management Centre, Universiti Teknologi MARA (REC/07/2021 (FB/44)).

### 2.2. Participants Recruitment

A convenience sampling method was applied, where children aged 9 to 16 years who attended the Faculty of Dentistry, Universiti Teknologi MARA (UiTM), along with their accompanying siblings, were invited to participate in this study. The inclusion criteria required children to have the ability to read and understand the Malay language and to have parental consent. Children with low literacy skills and mental disabilities were excluded. Prior to data collection of the validation phase, the minimum sample size required for psychometric testing of the Malay language COHIP-SF 19 was calculated. According to the “rule of thumb”, the ratio of the number of participants (N) to the number of measured variables (*p*) must be considered, with a recommended N ratio of 10 cases per indicator variable [20]. Thus, the minimum sample size required for confirmatory factor analysis (CFA) is at least 250 children [21]. To account for a potential 20% attrition rate, the target sample size was set at 300 children.

An information sheet was provided both to children who met the inclusion criteria and their parents or caregivers. Informed consent was obtained from the parents or caregivers, and assent was attained from the children who agreed to participate in this study. Participants completed a set of questionnaires that consisted of sociodemographic details including ethnicity and household income, dental visit history, and oral care habits, followed by the final version ML COHIP-SF 19. All research data were handled with strict confidentiality. Participants were anonymized and only unique identification numbers were used during data analyses. Data collection was carried out between January 2022 and April 2023.

### 2.3. Original English Version of the COHIP-SF 19 Questionnaire

The original COHIP-SF 19 in English was obtained from the developer [11]. It included 19 items, and 1 self-rated question (Q20) related to overall oral health perception, with response options ranging from 0 (poor) to 4 (excellent). Children reported how frequently they had experienced oral health impacts in the past 3 months using a five-point Likert scale, ranging from 0 (almost all the time) to 4 (never). The scoring for the two positively worded questions was reversed. The overall COHIP-SF 19 score was calculated by summing all 19 item scores within a range from 0 to 76. Higher COHIP-SF 19 scores reflected a better OHRQoL.

### 2.4. Translation and Pilot Test of the Prefinal Version of the Malay Language (ML) COHIP-SF 19

The translation and adaptation of the questionnaire followed an established forward- and back-translation method, adapted from Guillemin et al., 1993 [22], and Beaton et al., 2000 [23]. The forward translation was independently conducted by a pediatric dentistry postgraduate candidate (NA) and a professional translator from the Academy of Language Studies (PZ), both fluent in the Malay language and English. The translations were then reviewed, assessed, and revised by all authors to create a consensus version. Subsequently, the back-translation into English was performed by a second professional translator (EV), who was not familiar with the research area, and a public health postgraduate candidate (NB). All translation reports were generated and reviewed by an expert committee consisting of a pediatric dentistry specialist, a public health specialist, and a language expert. The reports were also reviewed by the questionnaire developer. Revisions were made to the translated questionnaire based on feedback from the expert committee and the questionnaire developer, to produce the prefinal version of the ML COHIP-SF 19. The content validity index (CVI) was calculated based on the relevant feedback provided by the expert committee members.

A pilot testing of the prefinal version of the ML COHIP-SF 19 questionnaire was conducted with 10 children aged from 9 to 16 years who were not included in the psychometric testing. Feedback gathered from the children focused on their comprehension of each question, including the meaning, clarity of wording, and relevance to oral health and its conceptual subscale, as well as their response options. The face validity index (FVI) was calculated based on the comments and feedback on the clarity of the questionnaire from ten raters during cognitive debriefing. It was observed that providing a picture for Item 3 (“Had discoloured teeth or spots on your teeth”) in the questionnaire improved participants’ understanding. Consequently, the final version of the ML COHIP-SF 19 was modified based on the feedback received from the participants, incorporating an intraoral photo which illustrates enamel opacities on the tooth surface to enhance the comprehension of Item 3 Appendix A.

### 2.5. Statistical Analysis: Psychometric Testing of the Final Version of the ML COHIP-SF 19

During the psychometric testing of the final version of the ML COHIP-SF 19, Cronbach’s alpha, the non-parametric Spearman’s correlation test, and confirmatory factor analysis (CFA) were used. Participants with more than 25% missing responses were excluded from data analyses. The internal consistency reliability for the overall scale and each subscale (oral health, functional well-being, and socio-emotional well-being) was determined using Cronbach’s alpha coefficient, with a coefficient of ≥0.7 considered satisfactory [24]. Since the data were not normally distributed, the non-parametric Spearman’s correlation test was used to evaluate convergent validity in determining the relationships between the ML-COHIP-SF 19 scores and self-perceived oral health ratings. In addition, confirmatory factor analyses (CFAs) were performed to confirm the factor structure of the questionnaire.

Absolute and incremental fit indices were used to evaluate the model fit. A two-index presentation format that includes SRMR with the NNFI (TLI), RMSEA, or CFI was suggested [25]. Others strongly advocate for the use of the chi-square test, RMSEA, CFI, and SRMR [26]. Values for an acceptable fit include a root mean square error of approximation (RMSEA) < 0.08, a comparative fit index (CFI) > 0.90, a Tucker–Lewis index (TLI) > 0.90, CMIN/DF < 3, and a standardized root mean squared residual (SRMR) < 0.08 [27]. All data analyses were conducted using SPSS software (IBM SPSS Statistics for Windows, Version 25.0. Armonk, NY, USA: IBM Corp.) and AMOS software (Amos 26.0 User’s Guide. Chicago, IL, USA: IBM SPSS. Version 28) [28]. Statistical significance was set at *p* < 0.05.

## 3. Results

### 3.1. Sociodemographic Data of Participants

A total of 252 children aged from 9 to 16 years were included in the final data analysis of this study. The mean age was 11.3 years (±1.87), and there were slightly more female participants (54.4%) than male. Most participants were Malay (98.8%), with 42.5% of the children coming from middle-income households with a monthly income in Ringgit Malaysia (RM) between RM 4850–RM 10,959. Additionally, 53.6% of the children’s parents had completed their tertiary education. The sociodemographic characteristics of the children recruited during the validation of ML COHIP-SF 19 are shown in Table 1.

### 3.2. Descriptive Analysis of the ML COHIP-SF 19

Descriptive data for the ML COHIP-SF 19 are presented in Table 2. The overall ML COHIP-SF 19 scores ranged from 20 to 75, with a mean score of 55.67 ± 10.45. No floor or ceiling effects were found for the total score. The ML COHIP-SF 19 scores by sex and age were examined using the Mann–Whitney test, and the data are presented in Table 3. Although males demonstrated slightly better OHRQoL scores than females, there were no significant differences in either the overall ML COHIP-SF 19 score or the subscale score for oral health according to sex. Children in the older age group had significantly lower social–emotional well-being scores (*p* = 0.04) compared to the younger age group.

### 3.3. Internal Consistency of ML COHIP-SF 19

The Cronbach’s alpha value (α) for the total ML COHIP-SF 19 score was 0.81, and the values for the three subscales were 0.60 for oral health (OH), 0.53 for functional well-being (FWB), and 0.76 for socio-emotional well-being (SEWB), as shown in Table 4. Items 15 and 18 showed the lowest item–rest correlation, which was below the recommended ideal value of above 0.3 (29). The overall Cronbach’s alpha of socio-emotional well-being improved slightly from 0.76 to 0.77 if Item 15 (“Felt that you were good looking”) was deleted, whereas the overall Cronbach’s alpha of functional well-being improved slightly from 0.53 to 0.54 if Item 18 (“Had difficulty keeping your teeth clean”) was deleted. However, both items were retained in the ML COHIP-SF 19.

### 3.4. Convergent Validity

Table 5 shows the convergent validity of the ML COHIP-SF 19. The average self-perceived oral health rating was 2.25 (±0.97). There was a significant correlation between the total ML COHIP-SF 19 and all the subscale scores with the perceived oral health ratings among the participants (*p* < 0.05). The observed correlation coefficients were positive (rs = 0.38–0.42), suggesting that the self-perceived oral health rating had a fair correlation with the ML COHIP-SF 19 total score and socio-emotional well-being subscale score (<0.01). On the other hand, the self-perceived oral health rating had little or no correlation with the oral health and functional well-being subscale scores (rs = 0.17–0.25). Thus, participants who rated their oral health more positively tended to have a higher ML COHIP-SF 19 total score and a higher socio-emotional well-being subscale score.

### 3.5. Confirmatory Factor Analysis

The four-factor model emerged from the confirmatory factor analysis (CFA) and exploratory factor analysis (EFA) results of previous studies [14,16,18] evaluating the structure of the original model, where Item 8 and Item 15 were extracted as a new factor, as shown in Figure 1. This appeared appropriate because these items [8,15] belonged to an independent subscale known as “Self-image” in the long version COHIP-34 questionnaire. The four-factor model showed an acceptable model fit with acceptable measures of fit values; however, the values of CFI and TLI were slightly lower than the good fit values (CFI > 0.90, TLI > 0.90) [27], while others recommended a cutoff value close to 0.95 for TLI and CFI [25]. The comparison of the measures of fit values of the four-factor models using CFA with previous validation studies is shown in Table 6.

## 4. Discussion

Malaysia is a multiethnic and multicultural country with the Malay language as its official language. The Malay language plays a pivotal role as a language of knowledge and communication, acting as the unifying factor for Malaysia’s diverse population that consists of Malay, Chinese, Indian, and other individuals representing the other indigenous Bumiputra groups in Malaysia. The COHIP-SF 19 is a potential OHRQoL measurement tool that can be utilized among primary and secondary schoolchildren in Malaysia, and this questionnaire has been translated and adapted into different versions. Having a single OHRQoL measurement tool in different languages offers the advantage of enabling the comparison of responses with international studies.

Even though the Indonesian version of the COHIP-SF 19 and the Malay CPQ that have been translated and validated in Brunei are available, these questionnaires might not be applicable to children in Malaysia due to the differences in colloquialisms and pronunciations. Moreover, as the Malay Child-OIDP index was validated among children aged from 11 to 12 years old, its appropriateness for younger children remains uncertain. Therefore, the advantage of the ML COHIP-SF 19 lies in its applicability to a wide age range, as the questionnaire has been validated among primary and secondary schoolchildren aged from 9 to 16 years in Malaysia. The ML COHIP-SF 19 is a comprehensive tool with 19 items, which allows a broader scope for evaluating the oral health impacts among children and adolescents. Compared to the Malay Child-OIDP index, which contains only eight items focusing on oral impact across eight performance areas, the ML COHIP-SF 19 covers a wider range of health and quality of life dimensions. While the Malay Child-OIDP minimizes the respondent burden due to fewer items, the more extensive ML COHIP-SF 19 may offer richer insights into oral health issues. Short form questionnaires, like this one, aim to balance detail and respondent convenience, with the Malay Child-OIDP prioritizing brevity and the ML COHIP-SF 19 offering a broader evaluation. The results of this study demonstrated that the psychometric characteristics were satisfactory, which suggests that it is suitable for future use with children aged from 9 to 16 years. As reliability is concerned with the ability of an instrument to measure consistently, it can be estimated using Cronbach’s alpha [29]. The reported reliability of the ML COHIP-SF 19 (α = 0.81) was adequate, with the Cronbach alpha value almost equal to that of the original COHIP-SF 19 (α = 0.82) [11]. Similar to the previous reported translation studies [16,18], the socio-emotional well-being subscale had the highest score (α = 0.76) with moderate reliability in regard to the oral health (α = 0.60) and functional well-being (α = 0.53) subscales. There are different reports about the acceptable values of alpha, ranging from 0.70 to 0.95 [21,29]. However, these values are still within the acceptable reliability range, as a Cronbach’s alpha value between 0.50 to 0.70 is considered to represent moderate reliability [30]. The small number of items included in the oral health and functional well-being subscale, with five items for oral health and four items for functional well-being, may have contributed to the lower scores from these two subscales [29].

Given that Southeast Asia includes Indonesia and Malaysia, it is appropriate to assess the performance of the COHIP-SF 19 questionnaire in these culturally distinct communities. The overall mean score for the Indonesian version of the COHIP-SF 19 [19] was 57.8 ± 8.8. The Indonesian version demonstrated slightly higher reliability (α = 0.83), with the socio-emotional well-being subscale having the highest score (α = 0.73), while the oral health (α = 0.59) and functional well-being (α = 0.65) subscales showed moderate reliability. The self-perceived oral health ratings exhibited little to no correlation with the total COHIP-SF 19 scores and all domain subscale scores (rs = 0.14–0.27).

The current study identified a low item–rest correlation for Item 15, consistent with the finding from earlier studies conducted in Germany [13], Japan [16], and France [18]. These studies similarly noted that removing Item 15 would slightly improve the overall Cronbach’s alpha. Additionally, the CFA showed relatively low factor loadings for positively worded Items 8 and 15. The Japanese study suggested that the findings could be influenced by cultural norms emphasizing modesty and humility, which are highly valued their culture [16]. However, a similar trend was observed in studies conducted in Western cultural contexts [13,18], suggesting that factors beyond culture may influence children’s response to these items. This highlights the importance of caution when interpreting data from positively worded items, as they may not accurately capture children’s quality of life. The findings of this study imply that children who do not rate positively worded items highly may not necessarily have a lower quality of life.

Factor analysis is a statistical method used to identify underlying factors (or latent variables) that explain patterns of correlations among observed variables [31]. Confirmatory factor analysis (CFA) and exploratory factor analysis (EFA) are the two primary subcategories of factor analysis. In this study, CFA was performed instead of EFA, as the previous studies have demonstrated the number of constructs and which construct theories or models best fit the data. The three-factor model retains the same structure as the original COHIP-SF 19m but previous studies have shown that using a four-factor model in CFA provides better factor loadings for items that exhibited weak relationships in the three-factor model, with slightly improved fit values for the COHIP-SF 19 [14,16,18]. Thus, this study employed a CFA four-factor model, where items 8 and 15 were grouped in one domain named “self-image” (SI), reflecting their original categorization under the “self-image” subscale in the original COHIP-34 version.

Although there are no “golden rules” for assessing model fit, it is important to report a range of indices, since each reflects different aspects of model fit. In this study, the values for CMIN/DF, CFI, TLI, RMSEA, and standardized RMR were recorded. The values for CMIN/DF, RMSEA, and standardized RMR indicated an acceptable fit. The scores for TLI and CFI range from 0 to 1, with higher values indicating a better fit. Consistent with previous validation studies, the CFI value in this study (CFI = 0.82) was below the established cutoff value [16,18]. There have been no previous reports of TLI values in related validation studies.

This study had some limitations. It was a cross-sectional study that employed a convenience sampling method and had a relatively small sample size compared to other validations of the COHIP-SF 19. Participants were recruited from a small town in Selangor, with the samples predominantly consisting of Malay children. Consequently, the use of a convenience sample, where participants were recruited from a teaching university, introduces potential bias as it is not representative of the general population. The findings should be interpreted with caution, as they may not fully represent the broader population of children in Malaysia. Therefore, it is recommended that future studies should consider using larger sample sizes, including from multiple centers, and recruit participants from diverse ethnic backgrounds with proficiency in Malay language skills to improve the generalizability of findings to Malaysian children. Additionally, larger and more varied samples of literacy and cognitive abilities will provide a more nuanced examination of cultural issues and model fit indicators.

This research represents the first oral health-related quality of life (OHRQoL) questionnaire validated for a broad age range of children in the Malay language. While the original COHIP-SF 19 is applicable to a wider age group, spanning from 7 to 18 years old, the present study focused on validating the ML COHIP-SF 19 for children aged from 9 to 16 years. Our findings indicate that future studies should consider testing and modifying the ML COHIP-SF 19 to suit children aged from 7 to 8 years. Despite these limitations, our results suggest that the ML COHIP-SF 19 is a valid tool for measuring the oral health-related quality of life among children and adolescents in Malaysia. Additionally, the questionnaire can be effectively integrated into national oral health program and clinical screenings to assess and improve oral health outcomes for pediatric populations in Malaysia.

## 5. Conclusions

The four-factor model ML COHIP-SF 19 demonstrated satisfactory psychometric characteristics, as well as an acceptable model fit, indicating its feasibility for use with Malaysian children aged from 9 to 16 years.

## Figures and Tables

**Figure 1 healthcare-13-00257-f001:**
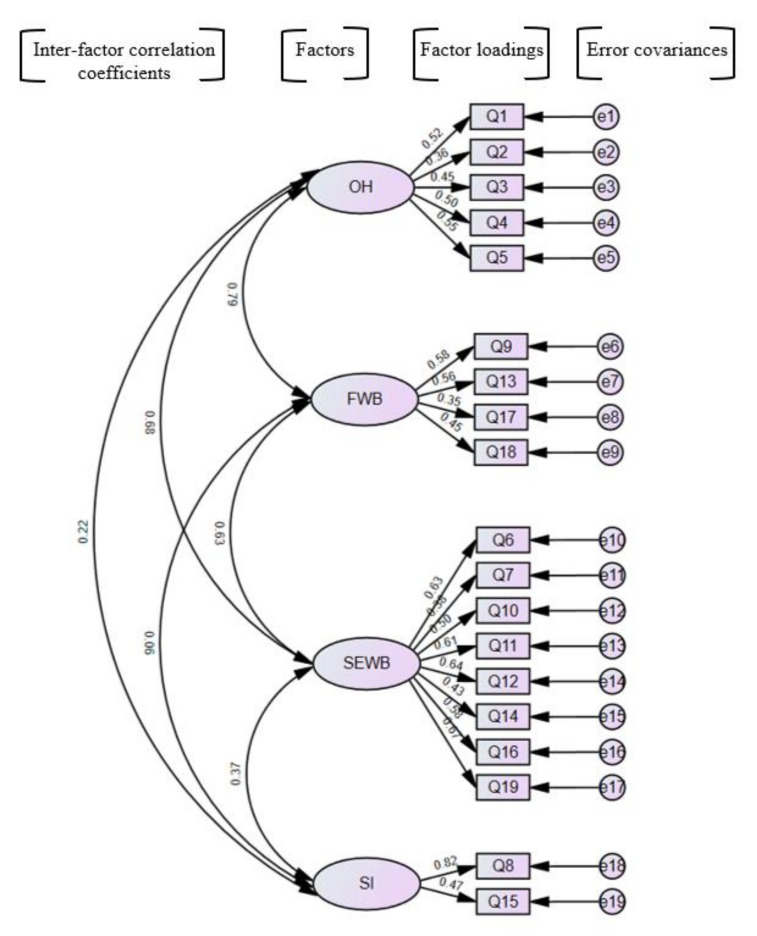
Four-factor model of the ML COHIP-SF 19 by confirmatory factor analysis. Note: OH: Oral Health, FWB: Functional Well-being, SEWB: Socio-emotional Well-being.

**Table 1 healthcare-13-00257-t001:** Sociodemographic characteristics of the children (*n* = 252).

Patient Characteristic		*n* (%)	Mean (±SD)
** *Child gender* **	Male	115 (45.6%)	
	Female	137 (54.4%)	
** *Ethnicity* **	Malay	249 (98.8%)	
	Indian	1 (0.4%)	
	Other	2 (0.8%)	
** *Household income* **	<RM 4850	69 (27.4%)	
	RM 4850–RM 10,959	107 (42.5%)	
	>RM 10,959	60 (23.8%)	
	No household income information	16(6.3%)	
** *Parental education* **	No formal education	2 (0.8%)	
	Primary education	-	
	Secondary education	94 (37.3%)	
	Tertiary education	135 (53.6%)	
	No parental education information	21 (8.3%)	

**Table 2 healthcare-13-00257-t002:** Descriptive data for overall and subscale scores of the ML COHIP-SF 19 (*n* = 252).

	Domain 1: Oral Health (*n* = 5)	Domain 2: Functional Well-Being (*n* = 4)	Domain 3: Socioemotional Well-Being (*n* = 10)	COHIP-SF 19 Total Score (*n* = 19)
**Mean score (±SD)**	13.63 (±3.92)	12.33 (±2.80)	29.70 (±6.36)	55.67 (±10.45)
**Range**	19 (1–20)	15 (1–16)	31 (9–40)	55 (20–75)
**Proportion of lowest possible score**	1 (0.4%)	1 (0.4%)	2 (0.8%)	1 (0.4%)
**Proportion of highest possible score**	9 (3.6%)	30 (11.9%)	7 (2.8%)	1 (0.4%)
**1st quartile**	11.0	11.0	26.0	49.0
**3rd quartile**	17.0	14.0	35.0	64.0

**Table 3 healthcare-13-00257-t003:** Descriptive data for overall and subscale scores of the ML COHIP-SF 19 by sex and age (*n* = 252).

	Sex			School		*p*-Value
	Male (*n* = 115)	Female (*n* = 137)		Primary School (9–12 Years Old) (*n* = 197)	Secondary School (13–16 Years Old) (*n* = 55)	
	Median (IQR)	Median (IQR)	*p*-Value	Median (IQR)	Median (IQR)	
**COHIP-SF 19 total score**	58.0 (15)	55.0 (18)	0.10	58.0 (16)	56.0 (19)	0.14
**Oral health**	14.0 (7)	13.0 (5)	0.07	14.0 (6)	13.0 (5)	0.32
**Functional well-being**	13.0 (4)	13.0 (3)	0.98	13.0 (4)	13.0 (3)	0.62
**Social–emotional well-being**	31.0 (8)	30.0 (9)	0.06	31.0 (9)	30.0 (10)	0.04 *

* indicates significant correlations (*p* < 0.05).

**Table 4 healthcare-13-00257-t004:** Internal reliability of the ML COHIP-SF 19 questionnaire (*n* = 252).

		Mean (SD)	Item–Rest Correlation	Total Cronbach’s Alpha If Item Deleted
**Domain 1: Oral health (OH)**				
Item 1	Mengalami sakit gigi *(Had pain in your teeth/toothache)*	2.93 (0.99)	0.26	0.59
Item 2	Mempunyai gigi yang tidak tersusun atau gigi jarang *(Had crooked teeth or spaces between your teeth)*	2.62 (1.48)	0.32	0.56
Item 3	Mempunyai gigi yang berbeza warna atau bertompok *(Had discolored teeth or spots on your teeth)*	2.70 (1.38)	0.39	0.52
Item 4	Mengalami masalah nafas berbau *(Had bad breath)*	2.57 (1.19)	0.40	0.52
Item 5	Mengalami masalah gusi berdarah *(Had bleeding gums)*	2.81 (1.22)	0.40	0.52
**Domain 2: Functional well-being (FWB)**				
Item 9	Susah untuk makan disebabkan oleh gigi, mulut atau muka anda *(Had difficulty eating foods I would like to eat)*	3.02 (1.14)	0.37	0.41
Item 13	Susah untuk tidur disebabkan oleh gigi, mulut atau muka anda *(Had trouble sleeping)*	3.49 (0.97)	0.44	0.37
Item 17	Susah untuk menyebut perkataan-perkataan tertentu *(Had difficulty saying certain words)*	3.48 (1.02)	0.26	0.51
Item 18	Susah untuk memastikan gigi anda sentiasa bersih *(Had difficulty keeping your teeth clean)*	2.35 (1.21)	0.24	0.54
**Domain 3: Socio-emotional well-being (SEWB)**				
Item 6	Berasa tidak gembira atau sedih disebabkan oleh gigi, mulut atau muka anda *(Been unhappy or sad)*	3.07 (1.20)	0.54	0.72
Item 7	Tidak hadir ke sekolah atas sebarang alasan yang berkaitan dengan gigi, mulut atau muka anda *(Missed school for any reason)*	3.61 (0.81)	0.33	0.75
Item 8	Berasa yakin (‘confident’) disebabkan oleh gigi, mulut atau muka anda *(Been confident)*	2.19 (1.33)	0.34	0.75
Item 10	Berasa bimbang atau risau disebabkan oleh gigi, mulut atau muka anda *(Felt worried or anxious)*	2.96 (1.09)	0.45	0.74
Item 11	Tidak mahu bercakap atau membaca di khalayak ramai di dalam kelas *(Not wanted to speak/read out loud in class)*	3.29 (1.06)	0.49	0.73
Item 12	Mengelak dari senyum atau ketawa bersama kanak-kanak lain disebabkan oleh gigi, mulut atau muka anda *(Avoided smiling or laughing with other children)*	3.33 (1.04)	0.53	0.73
Item 14	Diejek, dibuli atau digelar menggunakan panggilan nama yang tidak elok oleh kanak-kanak lain disebabkan oleh gigi, mulut atau muka anda *(Teased, bullied, or called names by other children)*	3.64 (0.89)	0.36	0.75
Item 15	Berasa diri anda menarik (cantik/kacak) disebabkan oleh gigi, mulut atau muka anda *(Felt that you were good looking)*	1.69 (1.36)	0.24	0.77
Item 16	Berasa diri kelihatan berbeza disebabkan oleh gigi, mulut atau muka anda *(Felt that you look different)*	3.00 (1.16)	0.46	0.74
Item 19	Berasa risau tentang pendapat orang lain terhadap gigi, mulut atau muka anda *(Been worried about what other people think)*	2.92 (1.24)	0.57	0.72

Note: The italics showed the English version of COHIP-SF 19.

**Table 5 healthcare-13-00257-t005:** Convergent validity of the self-perceived oral health rating with the ML COHIP-SF 19 score.

	Self-Perceived Oral Health	
	R	*p*-Value
Total ML COHIP-SF 19	0.38	<0.001
Oral health	0.25	<0.001
Functional well-being	0.17	<0.05
Socio-emotional well-being	0.42	<0.001

**Table 6 healthcare-13-00257-t006:** Comparison of the measures of fit values of the four-factor models using CFA in different validation studies.

Validation Study	χ^2^	DF	CMIN/DF (χ^2^/DF)	NFI	CFI	TLI	RMSEA	Standardized RMR
**ML COHIP-SF 19**	305.81	146	2.10	0.72	0.82	0.79	0.07	0.06
**COHIP-SF 19 JP** [16]	346.50	145	2.39	-	0.86	-	0.06	-
**French COHIP-SF-19** [18]	-	-	3.78	-	0.82	-	0.07	-

Note: DF: degree of freedom, RMSEA: root mean square error of approximation, NFI: Bentler–Bonett normed fit index, CFI: comparative fit index, TLI: Tucker–Lewis Index; values for acceptable fit: CMIN/DF < 3, NFI ≥ 0.95, CFI > 0.90, TLI > 0.90, RMSEA < 0.08, and SRMR < 0.09.

## Data Availability

The datasets used and/or analyzed during the current study and the Malay language COHIP-SF19 questionnaire are available from the corresponding author on reasonable request.

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
