# Peer review of "The Development and Psychometric Properties of Malay Language Child Oral Health Impact Profile—Short Form 19 (ML COHIP-SF 19)"

_healthcare, 2025, doi:10.3390/healthcare13030257_

Round 1
Reviewer 1 Report
Comments and Suggestions for Authors
The study design is well-structured, consisting of two phases with a clear focus on cross-cultural adaptation and validation of the COHIP-SF 19 questionnaire. However, the description of Phase 2, which involves evaluating the perceived impact of dental caries and dental pain, is mentioned but not detailed in this paper. Including a brief overview of Phase 2 could provide a more comprehensive understanding of the study's scope.
The inclusion criteria are appropriate, focusing on children who can read and understand Malay, but the exclusion of children with low literacy skills and mental disabilities might limit the generalizability of the findings.
The discussion section effectively summarizes the study's strengths and contributions, particularly the successful validation of the ML COHIP-SF 19. However, addressing the identified weaknesses and incorporating the suggested improvements would enhance the overall clarity and impact of the discussion. By emphasizing the need for larger, more diverse samples and providing a more nuanced exploration of cultural considerations and model fit indices, the discussion could offer a more comprehensive and insightful interpretation of the study's findings.
Author Response
Response to Reviewer 1 Comments |
|
|
|
Comments 1: The study design is well-structured, consisting of two phases with a clear focus on cross-cultural adaptation and validation of the COHIP-SF 19 questionnaire. However, the description of Phase 2, which involves evaluating the perceived impact of dental caries and dental pain, is mentioned but not detailed in this paper. Including a brief overview of Phase 2 could provide a more comprehensive understanding of the study's scope.
|
|
Response 1: Thank you for your comment. We have modified the paragraph and added” Phase 2 utilized the translated and validated questionnaire to evaluate the perceived impact of dental caries and dental pain on oral health-related quality of life (OHRQoL) of children in Malaysia. For Phase 2, dental caries was recorded using International Caries Detection and Assessment System (ICDAS) and dental pain was measured using Faces Pain Scales-Revised.’” Refer to line 88-92. We have also modified the introduction to emphasized the importance of Phase 2. “Addressing this gap is essential as it will provide more comprehensive understanding of Malaysian children's oral health. At present, no research has explored the impact of dental pain and dental caries on the OHRQoL of children in Malaysia.” Refer to line 74-77.
|
|
Comments 2: The inclusion criteria are appropriate, focusing on children who can read and understand Malay, but the exclusion of children with low literacy skills and mental disabilities might limit the generalizability of the findings.
Response 2: Thank you for pointing this out. We agree that this exclusion may limit the generalizability of the findings to the broader population, particularly those with diverse literacy and cognitive abilities. However, including children/adolescents with a wide range of literacy and cognitive abilities will introduce variables which may confound the results. We suggest future studies to consider a more inclusive approach to enhance the generalizability of the outcomes. This has been emphasized in the limitations of the study (line 330-335)
|
|
Comments 3: The discussion section effectively summarizes the study's strengths and contributions, particularly the successful validation of the ML COHIP-SF 19. However, addressing the identified weaknesses and incorporating the suggested improvements would enhance the overall clarity and impact of the discussion. By emphasizing the need for larger, more diverse samples and providing a more nuanced exploration of cultural considerations and model fit indices, the discussion could offer a more comprehensive and insightful interpretation of the study's findings.
|
|
|
Reviewer 2 Report
Comments and Suggestions for Authors
Thank you very much for the opportunity to review this manuscript. This study aimed to validate the Malay language COHIP-SF19 (ML COHIP-SF 19) and to assess its psychometric properties in Malaysian children/adolescents. This study is important for the journal readers, and I would like to congratulate the authors. I have a few suggestions and comments in order to improve this manuscript.
1. Keywords: Please review if they are MeSH termns
2. Please review if the manuscript accomplished the main rules for reporting observational studies (STROBE guidelines available in https://www.strobe-statement.org/) and Validation studies (i.e Hagströmer M, Ainsworth BE, Kwak L, Bowles HR. A checklist for evaluating the methodological quality of validation studies on self-report instruments for physical activity and sedentary behavior. J Phys Act Health. 2012 Jan;9 Suppl 1:S29-36. doi: 10.1123/jpah.9.s1.s29. PMID: 22287445)
3. Please justify if the sample is enough for validation studies.
4. Please mention recommendations for research and practice as derived from the main findings.
I hope these comments were useful for the authors and the best wishes for this manuscript.
Kind regards
Author Response
Comments 1: Thank you very much for the opportunity to review this manuscript. This study aimed to validate the Malay language COHIP-SF19 (ML COHIP-SF 19) and to assess its psychometric properties in Malaysian children/adolescents. This study is important for the journal readers, and I would like to congratulate the authors. I have a few suggestions and comments in order to improve this manuscript. Keywords: Please review if they are MeSH terms
Response 1: Thank you for your comment. We would like to clarify that these are keywords and not MeSH terms.
|
Comments 2: Please review if the manuscript accomplished the main rules for reporting observational studies (STROBE guidelines available in https://www.strobe-statement.org/) and Validation studies (i.e Hagströmer M, Ainsworth BE, Kwak L, Bowles HR. A checklist for evaluating the methodological quality of validation studies on self-report instruments for physical activity and sedentary behavior. J Phys Act Health. 2012 Jan;9 Suppl 1:S29-36. doi: 10.1123/jpah.9.s1.s29. PMID: 22287445)
|
Response 2: Thank you for pointing this out. Yes, this article adheres to the STROBE checklist guidelines for reporting cross-sectional studies.
|
Comments 3: Please justify if the sample is enough for validation studies |
Response 3: We would like to emphasize that the number of participants recruited represents the minimum sample size required for a validation study and confirmatory factor analysis. While we acknowledge that the sample size is smaller compared to other validation studies, it is sufficient to conduct the necessary analyses. Accordingly, we have revised the manuscript to include the following “Prior to data collection for the validation phase, the minimum sample size required for psychometric testing of the Malay language COHIP-SF 19 was calculated. According to the 'rule of thumb,' the ratio of the number of participants (N) to the number of measured variables (p) must be considered, with a recommended N ratio of 10 cases per indicator variable”. Refer line 106-110.
|
Comments 4: Please mention recommendations for research and practice as derived from the main findings.
|
Response 4: Thank you for your feedback. We mentioned in the limitation of the study that “Therefore, it is recommended that future studies should consider using larger sample sizes, including multi-centres, and recruit participants from diverse ethnic backgrounds with proficiency in Malay language skills to improve the generalizability of findings to Malaysian children. Additionally, larger and more varied samples of literacy and cognitive abilities will provide a more nuanced examination of cultural issues and model fit indicators. Refer line 330-335.
We also added the following statements “Our findings indicate that future study should consider testing and modifying ML COHIP-SF 19 to suit children aged 7 to 8 years. Despite these limitations, our results suggest that the ML COHIP-SF 19 is a valid tool for measuring the oral health related quality of life among children and adolescents in Malaysia. Additionally, the questionnaire can be effectively integrated into national oral health program and clinical screenings to assess and improve oral health outcomes for pediatric populations in Malaysia. Refer to line 340-345.
|
Reviewer 3 Report
Comments and Suggestions for Authors
The authors report the results of a study with 252 children aged 9 to 16 years in which the psychometric properties of a newly developed Malay language version of the Child Oral Health Impact Profile-Short Form 19 were tested. In summary the analyses presented here show good values for validity and reliability.
The paper is concise and well written. The results are of scientific interest. Tables are informative. The statistical procedures are appropriate.
Some suggestions:
Introduction:
+ please spend a few words on the Indonesian version of the questionnaire by Nuraini et al. 2021 and why a new developed Malay version provides an added value
2.4 2.4. Translation and pilot test
+ please provide the intraoral photo (item) 3 as picture or supplementary material
Results:
+ I recommend adding descriptive values for the total scale and the three domains for girls and boys. Are there differences in oral health?
+ I recommend adding descriptive values for the total scale and the three domains for (two or three?) different age groups. Are there differences in oral health?
+ table 1: It is not necessary to provide the ethnicity Chinese (zero participants)
+ table 1: I recommend to summarize the ethnicities Chinese, Indian, Other as “Other”
+ table 1 & line 175: please enter the full name for the abbreviation RM
+ table 1: please delete children’s age, the age is well described in text
+ table 3: please add more statistical information for the single items: mean, standard deviation, skewness, kurtosis
+ table 3: please delete Cronbach’s alpha in the table, the reliabilities are well described in text
+ 3.3., line 185 f.: I recommend to add McDonads Omega additionally
+ line 197: please name in the methods section how self-perceived oral health rating were measured (item wording, scaling)
+ table 4: I did not understand the abbreviation “rs (q)” for correlation. Why not just “r”?
+ table 4: I recommend adding the intercorrelation of the three subdomains of the questionnaire and the correlation of the subdomains with the total score as additional information on validity
+ table 5: I recommend adding df (degrees of freedom), NFI (Bentler-Bonett normed fit index), 90 % confidence interval (CI) for RMSEA
Discussion
+ please shorten, e.g. it is no necessary to repeat in which languages the questionnaire is translated (line 230, see line 65)
+ please explain in brief why a explanatory factor analysis were not provided
Author Response
Response to Reviewer 3 Comments |
Comments 1: The authors report the results of a study with 252 children aged 9 to 16 years in which the psychometric properties of a newly developed Malay language version of the Child Oral Health Impact Profile-Short Form 19 were tested. In summary the analyses presented here show good values for validity and reliability. The paper is concise and well written. The results are of scientific interest. Tables are informative. The statistical procedures are appropriate. Response 1: Thank for your feedback |
|
Comments 2: Introduction: i. Please spend a few words on the Indonesian version of the questionnaire by Nuraini et al. 2021 and why a new developed Malay version provides an added value ii. 2.4 2.4.Translation and pilot test Please provide the intraoral photo (item) 3 as picture or supplementary material
|
Responses 2: i. Agree. Therefore, we have explained this point that “even though the Indonesian version of COHIP-SF 19 is available, this questionnaire might not be applicable to children in Malaysia due to the difference in colloquialisms and pronunciation (line 68-71)”. ii. Thank you for pointing this out. We agree with this comment. Therefore, we have provided the intraoral photo (item) 3 as Supplementary material (Supplementary A)
|
Comments 3: Results i. I recommend adding descriptive values for the total scale and the three domains for girls and boys. Are there differences in oral health? i. Agree. We have added the data analysis on oral health according to gender to emphasize this point (Table 3). These statements added “The ML COHIP-SF 19 scores by sex and age were examined using the Mann Whitney test and the data are presented in Table 3. Although males demonstrated slightly better OHRQoL scores than females, there were no significant differences in either the overall ML COHIP-SF 19 score or the subscale score for oral health according to sex”. Refer line 193-197.
ii. I recommend adding descriptive values for the total scale and the three domains for (two or three?) different age groups. Are there differences in oral health? ii. Agree. We have, accordingly, done the data analysis on oral health according to age to address this comment (Table 3). This statement added “Children in the older age group had significantly lower social-emotional well-being scores (p = 0.04) compared to the younger age group”. Refer line 197-198.
iii. table 1: It is not necessary to provide the ethnicity Chinese (zero participants) iii. Agree. We have removed the ethnicity Chinese in table 1
iv. table 1: I recommend to summarize the ethnicities Chinese, Indian, Other as “Other” Thank you for pointing this out. We, however, will retain Indian and Other as individual item in the table as Malaysia is a multi-ethnic country where it is mainly consists of Malay, Chinese and Indian. Within Malaysian context, “Other” will represent the “other indigenous Bumiputra groups” in Malaysia (refer to line 245-249).
v. table 1 & line 175: please enter the full name for the abbreviation RM v. Thank you for pointing this out. We agree with this comment. Therefore, we have added the term “Ringgit Malaysia (RM)” in line 185 in the revised document.
vi. table 1: please delete children’s age, the age is well described in text vi. We have removed the children’s age in Table 1
vii. table 3: please add more statistical information for the single items: mean, standard deviation, skewness, kurtosis vii. We have added the mean and standard deviation for the single items in Table 3 (Table 4 in the revised document.
viii. table 3: please delete Cronbach’s alpha in the table, the reliabilities are well described in text viii. We have, accordingly, revised and removed the Cronbach’s alpha in the table 3 (Table 4 in the revised document)
ix. 3.3., line 185 f.: I recommend to add McDonads Omega additionally ix. Thank you for your suggestion, however for the current study we used Cronbach’s alpha values to measure the internal consistency and reliability of the items.
x. line 197: please name in the methods section how self-perceived oral health rating were measured (item wording, scaling) x. The self-perceived oral health rating measurement was explained in line 124-125. “one self-rated question (Q20) related to overall oral health perception, with response options ranging from 0 (poor) to 4 (excellent).”
xi. table 4: I did not understand the abbreviation “rs (q)” for correlation. Why not just “r”? xi. Agree. We have, accordingly modified the abbreviation “rs (q)” for correlation to “r” in table 4 (Table 5 in the revised document)
xii. table 4: I recommend adding the intercorrelation of the three subdomains of the questionnaire and the correlation of the subdomains with the total score as additional information on validity xii. Thank you for your suggestions. Previous studies that translated the COHIP-SF 19 into their languages did not include analyses of the intercorrelation among the three subdomains or their correlation with the total score as part of their validity testing. Therefore, these analyses were not conducted in our study. We believe the analyses performed are robust and sufficient to establish the validity of the questionnaire, aligning with the standard practices observed in similar study.
xiii. table 5: I recommend adding df (degrees of freedom), NFI (Bentler-Bonett normed fit index), 90 % confidence interval (CI) for RMSEA xiii. Agree. We have, accordingly revised the table and added df (degrees of freedom), NFI (Bentler-Bonett normed fit index) in Table 5 (Table 6 in the revised document)
|
Comment 4: Discussion i. please shorten, e.g. it is no necessary to repeat in which languages the questionnaire is translated (line 230, see line 65) ii. please explain in brief why a explanatory factor analysis were not provided
Response 4: i. Thank you for pointing this out. We agree with this comment. Therefore, we have modified and removed the repetition (originally line 230) ii. Thank you for pointing this out. We agree with this comment. Therefore, we have modified and explained on why EFA were not provided. “Factor analysis is a statistical method used to identify underlying factors (or latent variables) that explain patterns of correlations among observed variables [30]. Confirmatory factor analysis (CFA) and exploratory factor analysis (EFA) are the two primary subcategories of factor analysis. In this study, CFA was performed instead of EFA, as the previous studies have demonstrated the number of constructs, and which construct theories or models best fit.” Refer line 303-308 |